# Palladium and Platinum Complexes of the Antimetabolite Fludarabine with Vastly Enhanced Selectivity for Tumour over Non-Malignant Cells

**DOI:** 10.3390/molecules28135173

**Published:** 2023-07-02

**Authors:** Sebastian W. Schleser, Oleksandr Krytovych, Tim Ziegelmeier, Elisabeth Groß, Jana Kasparkova, Viktor Brabec, Thomas Weber, Rainer Schobert, Thomas Mueller

**Affiliations:** 1Organic Chemistry Laboratory, University Bayreuth, Universitaetsstrasse 30, 95447 Bayreuth, Germany; bt306701@uni-bayreuth.de (S.W.S.); oleksandr.krytovych@uni-bayreuth.de (O.K.); tim.ziegelmeier@tum.de (T.Z.); 2University Clinic for Internal Medicine IV, Hematology/Oncology, Medical Faculty, Martin Luther University Halle-Wittenberg, Ernst-Grube-Str. 40, 06120 Halle, Germany; elisabeth.gross@uk-halle.de (E.G.); thomas.weber@uk-halle.de (T.W.); thomas.mueller@medizin.uni-halle.de (T.M.); 3Department of Biophysics, Faculty of Science, Palacky University, Slechtitelu 27, 783 71 Olomouc, Czech Republic; jana@ibp.cz (J.K.); brabec@ibp.cz (V.B.)

**Keywords:** anticancer drugs, CLL, fludarabine, lymphoma, metal–drug synergy, platinum complexes

## Abstract

The purine derivative fludarabine is part of frontline therapy for chronic lymphocytic leukaemia (CLL). It has shown positive effects on solid tumours such as melanoma, breast, and colon carcinoma in clinical phase I studies. As the treatment of CLL cells with combinations of fludarabine and metal complexes of antitumoural natural products, e.g., illudin M ferrocene, has led to synergistically enhanced apoptosis, in this research study different complexes of fludarabine itself. Four complexes bearing a *trans*-[Br(PPh_3_)_2_]Pt/Pd fragment attached to atom C-8 via formal η^1^-sigma or η^2^-carbene bonds were synthesised in two or three steps without protecting polar groups on the arabinose or adenine. The platinum complexes were more cytotoxic than their palladium analogues, with low single-digit micromolar IC_50_ values against cells of various solid tumour entities, including cisplatin-resistant ones and certain B-cell lymphoma and CLL, presumably due to the ten-fold higher cellular uptake of the platinum complexes. However, the palladium complexes interacted more readily with isolated Calf thymus DNA. Interestingly, the platinum complexes showed vastly greater selectivity for cancer over non-malignant cells when compared with fludarabine.

## 1. Introduction

Leukaemia is among the ten most common causes of cancer mortality worldwide [1]. While acute lymphocytic leukaemia (ALL) is the most common type of cancer among children [2], chronic lymphocytic leukaemia (CLL) mainly affects the elderly. Therapy of the latter involves alkylating agents such as cyclophosphamide or purine analogues such as fludarabine (**1**) for physically fit patients, with antibody immunotherapies [3,4] as a costly alternative. Despite response rates of up to 90% and complete remission in 44% of previously untreated CLL patients [3,4], acquired resistance to fludarabine frequently thwarts its second-line administration [5]. Regimens combining these chemotherapeutics with antibodies have been shown in phase III clinical trials to be particularly efficacious [6]. Apart from lymphoproliferative malignancies, fludarabine has shown effects on various solid tumours such as melanoma, breast, and colon carcinoma [7]. Administered orally as a water-soluble monophosphate, fludarabine is rapidly dephosphorylated in the plasma, and is mono-, di-, and triphosphorylated after entering the cell [8]. As a purine analogue, it acts via incorporation into cellular DNA, ultimately inhibiting DNA synthesis and repair, transcription of RNA, and ribonucleotide reductase [9,10,11]. In addition, it is involved in mitochondrial pathways and in the induction of stress in the endoplasmic reticulum, modes of action which might allow fludarabine to circumvent cancer cell drug resistance [12]. Studies have shown that inherent or acquired resistance to treatment with platinum drugs may be overcome by combining them with fludarabine, a combination that inhibits the nucleotide excision repair (NER) machinery, with the effect of preserving DNA defects [13,14,15]. Previously, our group found that treatment of CLL cells with combinations of fludarabine and metal complexes, e.g., Illudin-M ferrocene, resulted in enhanced cytotoxicity [16]. This “metal–drug synergism” may be used to improve an already active drug’s cytotoxicity, stability, solubility, and pharmacokinetics [17]. There are examples of metallodrugs with peculiar modes of action, reduced unwanted side effects, or enhanced anti-tumoural activity against cancer cell lines that were originally refractory to the metal-free organic drug [18,19,20,21]. For the approach we intended here, namely, the application of metal complexes of fludarabine itself, there has been previous research. Leitão et al. reported the synthesis of Pd and Pt carbene complexes of the RNA constituent guanosine and their cytotoxicity against human cancer cell lines, with IC_50_ values in the high two-digit micromolecular range [22]. Herein, we report the synthesis of four palladium and platinum complexes of fludarabine bearing a *trans*-[Br(PPh_3_)_2_]Pt/Pd fragment attached to atom C-8 via η^1^-sigma or η^2^-carbene bonds, as well as their cytotoxicity against and selectivity for cancer cells of various solid tumour and lymphoma entities.

## 2. Results and Discussion

### 2.1. Complex Syntheses and Characterisation

The neutral *trans*-[(fludarabine–H)Br(PPh_3_)_2_]M complexes **3** (**a**: M = Pd; **b**: M = Pt) were synthesised in two steps following general protocols, with one additional reaction affording the corresponding cationic N-heterocyclic carbene (NHC) complexes **4** (Figure 1) [23]. The complexes were characterised by ^1^H, ^13^C, ^19^F, ^31^P, and ^195^Pt NMR spectra, ESI high-resolution mass spectrometry, and elemental analysis.

Fludarabine (**1**) was brominated selectively in the C-8 position, affording 8-bromofludarabine (**2**) in 66% yield. Unlike 8-chloropurines, bromide **2** underwent a smooth oxidative addition reaction with tetrakis(triphenylphosphane) complexes of Pd^0^ and Pt^0^, respectively, providing complexes **3a** and **b** in 63% yield. Several conceivable mechanisms for this oxidative addition were evaluated using DFT (density functional theory) calculations: a radical mechanism, an S_N_2-type reaction, and a concerted oxidative addition across the σ-bond between the halogen and carbon atom. The latter was considered most likely, and would result in a *cis*-configuration [24]. After oxidative addition, isomerisation can occur to provide the thermodynamically more stable *trans*-configured product [25]. Thus, the reaction was conducted at 80 °C overnight. As already pointed out by Leitão et al. for their related guanosine complexes, this step led to a mixture of compounds when using unprotected nucleosides [22]. Gratifyingly, the desired complexes **3**, aside from PPh_3_ and some Ph_3_PO, could be separated in a pure form by column chromatography. Palladium complex **3a** displayed two singlets in the ^31^P-NMR spectrum for the two phosphorus atoms, which are inequivalent due to the chiral ribosyl moiety [23]. The platinum complex **3b**, however, did not display such splitting. We assume that the two singlets coincide due to greater rotatability of the C–Pt bond. The fact that the singlets are not always nicely split or even visible at all is a phenomenon that has already been observed in previous works [23]. Yet, the platinum satellites in the ^31^P-NMR spectrum of **3b** were visible, with a *J* value of about 2750 Hz, which confirms the *trans* configuration of the complex. In accordance with this is the triplet found in the ^195^Pt-NMR spectrum with the same coupling constant. The observed coupling constants are in keeping with previously disclosed data of comparable complexes [26].

Complexes **3** were protonated with 2,6-lutidine triflate at the N-7 atom as the most basic position, yielding the cationic complexes **4** in 88%. Using a stronger acid, e.g., HBF_4_ · Et_2_O, led to decomposition of the complexes. A distinct roofing effect of the two phosphorus signals was visible in the ^31^P-NMR spectra of complexes **4a** and **4b,** with ^2^*J*_PP_ values of 454 Hz and 418 Hz, respectively, in accordance with the previous literature on comparable complexes [23]. Platinum complex **4b** showed satellites with *J*_Pt-P_ = 2500 Hz in its ^31^P-NMR spectrum and a triplet with the same coupling constant in the ^195^Pt-NMR spectrum. The complexes showed no change of their signals in ^1^H-NMR spectra (cf. Appendix A) over a period of at least three days when dissolved in DMSO-d_6_ + 5% D_2_O, i.e., under conditions used in the bioevaluation assays, and as such can be considered stable and causative for the biological effects. The isotope exchange with a large excess of D_2_O can explain the disappearance of the amine and hydroxy protons immediately after adding water.

### 2.2. Cytotoxicity

The complexes were investigated for their cytotoxicity against panels of solid tumour, lymphoma, and leukaemia cell lines. To assess activity in models of solid tumours, compounds were analysed using the SRB cytotoxicity assay, employing a selection of human tumour cell lines of different entities and non-malignant human fibroblasts (CCD18Co) to assess both their anti-proliferative/cytotoxicity activity and their tumour cell/non-malignant cell selectivity. In addition, with the cell line pair A2780/A2780cis, we used a panel comprising a known model of acquired drug resistance to conventional drugs such as cisplatin and doxorubicin. The results are compiled in Table 1.

The upshot of these studies was that none of the complexes is more cytotoxic against any of the tumour cell lines than fludarabine, with the platinum complexes **b** being more active than the palladium complexes **a**. Most remarkable is the fact that all complexes, in particular platinum complex **4b**, are distinctly more cytotoxic against tumour cells than non-malignant cells, unlike fludarabine. 

Next, complexes **3** and **4** were tested against a panel of one human B-cell and two human T-cell lymphoma cell lines as well as a chronic lymphocytic leukaemia cell line. Again, **4b** was the most active of the four complexes, reaching almost 50% of the activity of fludarabine (**1**) against human large B-cell lymphoma Oci-Ly1 and the human CLL HG-3 cells (Table 2).

### 2.3. Cellular Uptake and DNA Metalation

To clarify whether the cytotoxicities of complexes **3** and **4** are intrinsic and structure-dependent or a consequence of the respective cellular concentration, we assessed their cellular uptake by HCT116 colon carcinoma cells as the extent of the metalation of the cellular DNA via ICP-MS. To this end, we first measured their cytotoxicity by means of MTT assays (Table 3).

Again, the platinum complexes **3b** and **4b** were more active than their palladium analogues **3a** and **4a** by a factor of ten, with little difference within the couples of neutral and cationic complexes sharing the same metal, i.e., **3a**<=>**4a** and **3b**<=>**4b**. The role of the platinum fragment becomes clear upon examining the cellular uptake and DNA interactions of the complexes (Figure 1).

The proportion of metal taken up correlates indirectly with the IC_50_ values of the complexes. The uptakes of platinum complexes **3b** and **4b** of 645 ± 95.0 and 784 ± 64.0 pmol/10^6^ cells, respectively, are increased by a factor of ten relative to the palladium complexes **a**, as are the cytotoxicities. In both cases, the cationic complexes **4** are taken up to a larger extent compared to the neutral complexes **3**. A possible explanation for the lower uptake of **3a** and **4a** may be the higher reactivity of palladium complexes, which might lead to unwanted side reactions that hinder entrance into the cell, eventually rendering the complexes inactive before reaching their target [28]. 

However, the percentual DNA metalation relative to the overall cellular uptake, displayed as the second *y*-axis and grey columns in Figure 1, was higher for the palladium complexes **a** than for the platinum complexes **b**, perhaps following the same rationale.

### 2.4. DNA Binding in Cell-Free Medium

The latter statement was proven in experiments with isolated calf thymus DNA. The results are shown in Figure 2. After 24 h, about 40% of the added palladium complexes **3a** and **4a** were bound to this DNA, while the platinum analogues **3b** and **4b** were only 15% and 17% bound, respectively. Cisplatin, used as a positive control, was bound completely after 24 h. This is proof that the palladium complexes **a** of fludarabine are more “reactive” towards DNA compared to the platinum complexes **b**. This is likely due to steric reasons, such as the bulkiness and flexibility of the respective complexes, rather than to the metals as such, as is apparent from the high “reactivity” of cisplatin.

## 3. Materials and Methods

### 3.1. General

Elemental analyses were carried out with a Perkin-Elmer (Waltham, MA, USA) 2400 CHN elemental analyser. 

Melting points were taken with an Electrothermal 9100 apparatus and are uncorrected. 

HR-ESIMS (High-resolution electrospray ionization mass spectrometry) spectra were recorded with a Thermo Fisher Scientific (Waltham, MA, USA) UPLC/Orbitrap MS system.

Nuclear magnetic resonance (NMR) spectra were measured using a Bruker (Billerica, MA, USA) DRX spectrometer at ambient temperature. Chemical shifts are provided in parts per million (*δ*) downfield from tetramethylsilane as an internal standard. As an internal standard for ^1^H-NMR spectra, the resonance signal of the residual protons of CD_3_CN (*δ* = 1.94 ppm), DMSO-d_6_ (*δ* = 2.50 ppm) or CD_2_Cl_2_ (*δ* = 5.32 ppm) was used. For ^13^C-NMR spectra, the resonance signal of the carbon atom of CD_3_CN (*δ* = 118.3 ppm), DMSO-d_6_ (*δ* = 39.5 ppm) or CD_2_Cl_2_ (*δ* = 53.8 ppm) was used [23]. The ^1^H-NMR spectra were measured at 500 MHz, ^13^C-NMR spectra at 125 MHz, ^19^F-NMR spectra at 470 MHz, ^31^P-NMR spectra at 202 MHz, and ^195^Pt-NMR spectra at 107 MHz. For signal multiplicities, the following abbreviations were used: s = singlet, d = doublet, t = triplet, m = multiplet, dd = doublet of doublets, dt = doublet of triplets. Coupling constants are provided in Hz. 

For purification of synthetic products, chromatography silica gel 60 (40–63 μm) or silica gel RP18 (40–63 μm) was used. Analytical thin layer chromatography (TLC) was carried out using Merck silica gel 60 F_254_ pre-coated aluminium-backed plates. 

Starting compounds were purchased from Sigma-Aldrich (St. Louis, MI, USA), TCI (Tokyo, Japan), Merck (Darmstadt, Germany), abcr (Karlsruhe, Germany), Acros Organics (Fair Lawn, NJ, USA), and VWR (Radnor, PA, USA), and used without further purification. All reactions with moisture-sensitive reagents were carried out under an argon atmosphere in water-free solvents. Unless stated otherwise, the solvents were purified and dried using standard methods.

### 3.2. Chemistry

#### 3.2.1. 8-Bromofludarabine (**2**)

Analogously to the literature [23], fludarabine (300 mg, 1.05 mmol, 1.00 eq.) was treated with 27 mL of 0.5 mm NaOAc buffer (pH = 4, 1.11 g, 13.5 mmol, 12.8 eq.); 50 mL of a saturated aqueous solution of bromine was added dropwise, and the resulting mixture was stirred at room temperature for three days. Na_2_S_2_O_5(aq.)_ was added until the colour almost disappeared. 2M NaOH was added until a pH value of 7 was reached. The mixture was filtered and the solid residue was washed with water and acetone and dried to leave **2** (244 mg, 670 µmol, 66%) as a white solid of m.p. 185 °C (decomp.). R_f_ = 0.76 (CH_2_Cl_2_: MeOH 7:3); ^1^H-NMR (500 MHz, DMSO-d_6_): δ = 7.96 (d, *J* = 68.1 Hz, 2H, NH_2_), 6.17 (d, *J* = 6.6 Hz, 1H), 5.57 (d, *J* = 5.5 Hz, 1H), 5.47 (d, *J* = 5.2 Hz, 1H), 4.82 (dd, *J* = 6.3 Hz, 4.7 Hz, 1H), 4.39–4.29 (m, 2H), 3.81 (dt, *J* = 12.0 Hz, 6.8 Hz, 1H), 3.75–3.68 (m, 2H) ppm; ^19^F-NMR (470 MHz, DMSO-d_6_): δ = −51.23 (s) ppm; ^13^C-NMR (125 MHz, DMSO-d_6_): δ = 158.3 (d, *J*_C–F_ = 204 Hz), 157.0 (d, *J*_C–F_ = 21.5 Hz), 151.9 (d, *J*_C–F_ = 20.3 Hz), 126.1 (s), 118.1 (d, J_C–F_ = 3.60 Hz), 86.3 (s), 83.4 (s), 76.7 (s), 74.9 (s), 61.8 (s) ppm.

#### 3.2.2. General Procedure for the Oxidative Addition of **2**

A solution of **2** (1.00 eq.) in dry THF (200 mL/mmol) kept under an argon atmosphere was treated with Pd(PPh_3_)_4_ (1.00 eq.) or Pt(PPh_3_)_4_ (1.00 eq.), then the resulting mixture was heated to 80 °C for 16 h in a tightly sealed flask. The solvent was evaporated and the residue was purified via column chromatography, affording **3** as a yellowish solid after drying in vacuo.

#### 3.2.3. *trans*-[bromido-(fludarabin-8-yl)-bis(triphenylphosphane)]palladium(II) (**3a**)

174 mg (175 µmol, 63%) from **2** (100 mg, 275 µmol) and Pd(PPh_3_)_4_ (317 mg, 275 µmol) in dry THF (55 mL). R_f_ = 0.46 (CH_2_Cl_2_: MeOH 95:5); m.p. 250 °C; ^1^H-NMR (500 MHz, CD_3_CN): δ = 7.65 (dd, *J* = 4.6 Hz, 2.3 Hz, 6H), 7.56 (q, *J* = 5.9 Hz, 6H), 7.46–7.16 (m, 18H), 6.53 (dd, *J* = 13.2 Hz, 3.2 Hz, 1H), 5.73 (s, 2H, NH_2_), 5.41 (dd, *J* = 9.2 Hz, 7.3 Hz, 1H), 4.53–4.35 (m, 1H), 4.07 (dt, *J* = 13.4 Hz, 1.8 Hz, 1H), 3.79–3.70 (m, 1H), 3.66 (dd, *J* = 24.4 Hz, 3.3 Hz, 1H), 3.48–3.34 (m, 3H) ppm; ^13^C-NMR (125 MHz, DMSO-d_6_): 160.9 (s), 154.2 (d, *J*_C–F_ = 204 Hz), 154.1 (d, *J*_C–F_ = 20.5 Hz), 151.3 (d, *J*_C–F_ = 18.3 Hz), 134.4 (m), 130.6 (m), 130.2 (m), 128.0 (m), 121.6 (s), 88.3 (s), 84.61 (s), 76.0 (s), 75.8 (s), 61.43 (s) ppm; ^19^F-NMR (470 MHz, CD_3_CN): δ = −57.90 (s) ppm; ^31^P-NMR (202 MHz, CD_3_CN): δ = 21.7 (s), 21.2 (s) ppm; HRMS (ESI): *m*/*z* calcd. for C_46_H_41_BrFN_5_O_4_P_2_Pd + H^+^: 994.0909 [*M* + H]^+^; found: 994.0897; elemental analysis calcd. (%): C 55.52, H 4.15, N 7.04; found: C 56.22, H 4.30, N 6.98.

#### 3.2.4. *trans*-[bromido-(fludarabin-8-yl)-bis(triphenylphosphane)]platinum(II) (**3b**)

181 mg (167 µmol, 63%) from **2** (98.0 mg, 269 µmol) and Pt(PPh_3_)_4_ (335 mg, 269 µmol). R_f_ = 0.50 (CH_2_Cl_2_: MeOH 95:5); m.p. 290 °C; ^1^H-NMR (500 MHz, CD_3_CN): δ = 7.65 (dd, *J* = 34.6 Hz, 7.32 Hz, 12H), 7.42–7.27 (m, 18H), 6.87 (d, *J* = 2.9 Hz, 1H), 5.95 (s, 2H, NH_2_), 5.27 (d, *J* = 8.3 Hz, 1H), 4.80 (s, 1H), 4.08 (s, 1H), 3.79 (s, 1H), 3.64 (s, 1H), 3.54–3.39 (m, 3H) ppm; ^13^C-NMR (125 MHz, CD_2_Cl_2_): δ = 153.8 (d, *J*_C–F_ = 207 Hz), 151.7 (d, *J*_C–F_ = 20.4 Hz), 150.2 (d, *J*_C–F_ = 19.5 Hz), 149.1 (s), 132.9 (d, *J*_C–P_ = 6.60 Hz), 131.2 (d, *J*_C–P_ = 32.2 Hz), 129.0 (d, *J*_C–P_ = 32.1 Hz), 128.0–127.5 (m), 126.3 (dt, *J*_C–P_ = 29.8 Hz, *J*_C–Pt_ = 5.30 Hz), 89.1 (s), 83.4 (s), 77.5 (s), 75.7 (s), 60.7 (s) ppm; ^19^F-NMR (470 MHz, CD_2_Cl_2_): δ = −57.24 (s) ppm; ^31^P-NMR (202 MHz, CD_2_Cl_2_): δ = 18.7 (s) ppm, Pt satellites *J*_P–Pt_ = 2744 Hz; ^195^Pt-NMR (107 MHz, CD_2_Cl_2_): δ = −4496 (t, *J*_Pt–P_ = 2744 Hz) ppm; HRMS (ESI): *m*/*z* calcd. for C_46_H_41_BrFN_5_O_4_P_2_Pt + H^+^: 1083.1522 [*M* + H]^+^; found: 1083.1508; elemental analysis calcd. (%): C 50.98, H 3.81, N 6.46; found: C 50.53, H 3.78, N 6.10.

#### 3.2.5. General Procedure for N7-Protonation of Complexes **3**

A solution of **3** (1.00 eq.) in dry CH_2_Cl_2_ (200 mL/mmol) under argon atmosphere was treated with 2,6-lutidine triflate (1.00 eq.), then the resulting mixture was stirred at room temperature for 24 h. The solvent was evaporated and the residue was washed with diethyl ether, cooled to −20 °C, and dried in vacuo to afford cationic complex **4** as a yellowish solid.

#### 3.2.6. *trans*-[bromido-(7-H-fludarabin-8-ylidene)-bis(triphenylphosphane)]palladium(II) triflate (**4a**)

92.0 mg (80.3 µmol, 88%) from **3a** (91.0 mg, 91.4 µmol) and 2,6-lutidine triflate (23.5 mg, 91.4 µmol). m.p. 190 °C; ^1^H-NMR (500 MHz, DMSO-d_6_): δ = 12.2 (s, NH^+^), 7.70–7.56 (m, 11H), 7.54–7.36 (m, 19H), 5.67–5.63 (m, 2H), 5.56–5.40 (m, 1H), 4.26 (q, *J* = 4.2 Hz, 1H), 4.21–4.11 (m, 1H), 3.77 (dt, *J* = 6.1 Hz, 3.5 Hz, 1H), 3.66 (dt, *J* = 11.3 Hz, 5.5 Hz, 1H), 3.45–3.16 (m, 4H) ppm; ^13^C-NMR (125 MHz, DMSO-d_6_): 167.8 (s, OTf), 155.6 (d, *J*_C–F_ = 207 Hz), 149.2 (d, *J*_C–F_ = 21.3 Hz), 147.9 (d, *J*_C–F_ = 20.4 Hz), 132.1 (s), 129.3 (d, *J*_C–F_ = 25.4 Hz), 126.6 (dd, *J*_C–F_ = 24.8 Hz, 7.9 Hz), 118.8 (q, *J*_C–F_ = 322.4 Hz, OTf), 109.0 (s), 87.3 (s), 84.1 (s), 74.3 (s), 74.0 (s), 59.9 (s) ppm; ^31^P-NMR (202 MHz, DMSO-d_6_): δ = 19.5 (s), 19.2 (s) ppm; HRMS (ESI, pos): *m*/*z* calcd. for C_46_H_42_BrFN_5_O_4_P_2_Pd^+^: 994.0909 [*M*-OTf]^+^; found: 994.0909, HRMS (ESI, neg): *m*/*z* calcd. for CF_3_O_3_S^−^ 148.9526; found: 148.9509; elemental analysis calcd. (%): C 49.29, H 3.70, N 6.12; found: C 49.20, H 3.90, N 5.89.

#### 3.2.7. *trans*-[bromido-(7-H-fludarabin-8-ylidene)-bis(triphenylphosphane)]platinum(II) triflate (**4b**)

110 mg (89.2 µmol, 88%) from **3b** (110 mg, 102 µmol) and 2,6-lutidine triflate (26.1 mg, 102 µmol). m.p. 230 °C; ^1^H-NMR (500 MHz, DMSO-d_6_): δ = 12.1 (s, NH^+^), 7.70–7.55 (m, 11H), 7.53–7.34 (m, 19H), 6.68 (d, *J* = 5.8 Hz, 1H), 5.45 (s, 2H), 4.57 (s, 1H), 4.30 (s, 1H), 4.05 (s, 1H), 3.57 (q, *J* = 5.2 Hz, 1H), 3.50–3.44 (m, 2H), 3.34 (s, 2H) ppm; ^13^C-NMR (125 MHz, DMSO-d_6_): 157.9 (d, *J*_C–F_ = 207 Hz), 151.7 (d, *J*_C–F_ = 20.9 Hz), 150.2 (d, *J*_C–F_ = 20.4 Hz), 134.6 (s), 131.9 (d, *J*_C–F_ = 30.4 Hz), 129.1 (dd, *J*_C–F_ = 35.9 Hz, 8.2 Hz), 121.1 (q, *J*_C–F_ = 322.4 Hz, OTf), 110.8 (s), 90.0 (s), 85.3 (s), 77.1 (s), 76.3 (s), 65.4 (s) 62.0 (s) ppm; ^31^P-NMR (202 MHz, DMSO-d_6_): δ = 16.7 (s), 16.3 (s) ppm, Pt satellites *J*_P–Pt_ = 2503 Hz; ^195^Pt-NMR (107 MHz, DMSO-d_6_): δ = −4451.7 (t, *J*_Pt–P_ = 2503 Hz) ppm; HRMS (ESI): *m*/*z* calcd. for C_46_H_42_BrFN_5_O_4_P_2_Pt^+^: 1083.1522 [*M* + H]^+^; found: 1083.1472; HRMS (ESI, neg): *m*/*z* calcd. for CF_3_O_3_S^−^ 148.9526; found: 148.9509; elemental analysis calcd. (%): C 45.75, H 3.43, N 5.02; found: C 46.05, H 3.63, N 5.49.

### 3.3. Bioevaluation

#### 3.3.1. Cell Culture

The human cancer cell lines A2780 (ECACC #93112519), A2780Ccis (ECACC # 93112517), A549 (ATCC-CCL-185), HT29 (ATCC-HTB-38), and MCF7 (ATCC-HTB-22) were cultivated in RPMI-1640 medium. Non-malignant human fibroblasts CCD18Co (ATCC-CRL-1459) were grown in MEME (both from Sigma-Aldrich, St. Louis, MO, USA). Both media were supplemented with 10% fetal bovine serum (Biowest, Nuaillé, France) and 1% penicillin-streptomycin (Sigma-Aldrich).

The cell lines HH (Cat# ACC 707; RRID:CVCL_1280), and HG-3 (Cat# ACC 765; RRID:CVCL_Y547) were obtained from the German Collection of Microorganisms and Cell Cultures (DSMZ, Braunschweig, Germany) and cultivated in RPMI-1640 Medium supplemented with 10% fetal bovine serum and 1% penicillin/streptavidin (P0781, Sigma-Aldrich, St. Louis, MO). DERL-2 (Cat# ACC 531; RRID:CVCL_2016) was obtained from DSMZ and cultivated in RPMI-1640 Medium with 20% fetal bovine serum, 1% penicillin/streptavidin, and 0.02 ng/µL recombinant IL-2 (200-02, Peprotech, Hamburg, Germany). Oci-Ly1 (Cat# ACC 722; RRID:CVCL_1879) was obtained from DSMZ and cultivated in RPMI-1640 Medium with 20% fetal bovine serum and 1% penicillin/streptavidin.

All cell lines were cultured in a humidified atmosphere at 37 °C and 5% CO_2_. Passaging did not exceed ten times after thawing. Mycoplasma negativity was confirmed repeatedly using MycoSPY^®^ Master Mix (M020, Biontex, Munich, Germany).

#### 3.3.2. Resazurin Cell Viability Assay (Suspension Culture)

Cells were seeded at the indicated densities in 100 µL per well in eight replicates. Detailed densities are listed in Table 4.

Cells were treated with the compounds for 72 h in two serial dilutions ranging from 100 µM to 0.01 µM or 30 µM to 0.03 µM, respectively. Cell viability was measured indirectly by fluorescence detection with a Spark^®^ multimode microplate reader (Tecan, Männedorf, Switzerland) after 2 h of Resazurin treatment (30 µg/mL, R12204, Thermo Fisher Scientific, Waltham, MA). Each experiment was performed thrice, and GraphPad Prism software v8.4.3 (GraphPad, San Diego, CA, USA; RRID:SCR_002798) was used for calculation of individual and mean IC_50_ values.

#### 3.3.3. SRB Assay (Monolayer Culture)

Cytotoxic activities of compounds in models of solid tumours were analysed using the SRB cytotoxicity assay. Cells were seeded in 96-well plates, and after 24 h were treated with serial dilutions of the compounds (30 µM to 0.003 µM) for 72 h. All subsequent steps were performed according to the previously described SRB assay protocol [29]. Dose–response curves and calculation of IC_50_ values, including standard deviations, were carried out using GraphPad Prism8.

#### 3.3.4. Intracellular Accumulation

HCT 116 cells (1 × 10^6^) were seeded on 100 mm Petri dishes, incubated overnight, and then treated with the tested compounds at their 10 µM concentration for 6 h. The cells were then exhaustively washed with PBS, harvested by trypsinisation, counted, and washed twice with ice-cold PBS. The cell pellets were digested using the microwave acid (HCl, 5 M) digestion system (CEM Mars^®^). The quantity of platinum or palladium taken up by the cells was determined by inductively coupled plasma mass spectrometry (ICP-MS).

#### 3.3.5. Determination of the Amount of Metal Associated with DNA

HCT116 cells were seeded and treated with the test complexes as described above. The cell pellets were stored at −70 °C and then lysed in DNAzol (DNAzol, MRC) supplemented with RNase A (100 mg mL^−1^). Genomic DNA was precipitated from the lysate using 100% ethanol, washed, and resuspended in 8 mM NaOH. The DNA content in each sample was determined by UV spectrophotometry. To avoid interference from high DNA concentrations when detecting metals in the samples, the DNA samples were digested in the presence of 30% hydrochloric acid (Suprapur, Merck Millipore, Burlington, MA, USA). The amount of metal bound to nucleic acids was quantified by ICP-MS.

#### 3.3.6. DNA Binding in Cell-Free Media

Solutions of double-helical calf thymus DNA (42% G + C) at a concentration of 32 μg mL^−1^ were incubated with the tested complexes in 10 mM NaClO_4_ at 37 °C. The molar ratio of metal complex to nucleotide residue was 0.2. After 24 h incubation, the reaction was stopped by adding NaCl (1 M) and the samples were quickly cooled in a dry ice bath. The samples were exhaustively dialysed against 0.1 M NaCl (1 h), and subsequently against water (2 × 2 h) to remove free unbound complexes. The concentrations of DNA and the content of metal associated with the DNA were determined by absorption spectrophotometry and FAAS (Varian AA240Z), respectively.

## 4. Conclusions

Four new water-stable palladium and platinum complexes, with the clinically established antimetabolite fludarabine as an η^1^-alkyl or η^2^-carbene ligand, were prepared in a few steps, in high yields, and without using protecting groups. This route should be applicable to other purine nucleoside-derived antimetabolites. While none of the complexes was more cytotoxic than fludarabine against any cancer cell line of the various tested solid tumour and lymphoma entities, the platinum complexes **3b** (neutral) and **4b** (cationic) surpassed their palladium analogues **3a** and **4a**, with clinically relevant low single-digit IC_50_ values for seven out of nine cancer cell lines. What should catch the interest of clinicians even more is the fact that the platinum complexes showed vastly greater selectivity for cancer over non-malignant cells when compared with the lead compound fludarabine. Mechanism-wise, DNA appears to be their main target, which is little wonder for Pt complexes of DNA base surrogates. A plausible assumption is that all complexes form structurally identical DNA adducts and that the difference in the overall cytotoxicity of the complexes stems predominantly from their different cellular accumulation. While the more cytotoxic platinum complexes experienced a roughly ten-fold higher uptake into cancer cells, the palladium complexes bound more readily or to a greater extent to isolated pure Calf thymus DNA.

## Data Availability

Data supporting the findings of this study are available from the corresponding author upon reasonable request.

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
