# Peer review of "Palladium and Platinum Complexes of the Antimetabolite Fludarabine with Vastly Enhanced Selectivity for Tumour over Non-Malignant Cells"

_molecules, 2023, doi:10.3390/molecules28135173_

Round 1
Reviewer 1 Report
The paper of Sebastian W. Schleser and co-authors is an interesting fundamental work on synthesis 4 new Platinum and Palladium complexes with biological active anti-cancer agent fludarabine. All complexes were studied using NMR on different nuclei. The article is written very well, neatly, I have no major remarks. However, there are a few questions and suggestions that I need to ask.
It is not entirely clear from the introduction and the text of the article why the authors synthesize a bromine derivative of platinum, while it has long been proven that in cisplatin it is the platinum-chlorine bond and its dynamic stability that plays a key role in anticancer activity. Was it possible to chlorinate the ligand at the first stage and obtain a chlorine derivative of platinum and palladium? Perhaps the authors would have obtained a synergistic effect from the presence of fludarabine and the dissociation of the platinum-chlorine bond.
It is very unfortunate that the authors failed to grow single crystals for any substance. It is possible that isomers with different biological activity may exist in the crystalline form. In solution, the existence of isomers may not be noticed since the dynamics in solution can lead to the prevalence of one form. That in vivo research can give rise to problems.
According to experimental data, the reaction takes place in dry tetrahydrofuran, in argon, and even in a sealed ampoule. Where does the impurity of triphenylphosphine oxide come from?
The presented manuscript is a very interesting work and I certainly recommend accepting it for publication in the Molecules. This work will definitely attract a lot of attention from researchers and I hope increase citations for the Molecules.
Reviewer 2 Report
The authors described palladium and platinum complexes of the antimetabolite fludarabine, and presented the in vitro celullar activities on several tumor cell lines and non-malignant human cell with the expectation of obtaining the enhanced selectivity for tumour over non-malignant cells. Although tthe manuscript presented some interesting results, I do NOT suggest to accept it based on several comments.
1. The fludarabine (1) was used for treatment of lymphocytic leukaemia, I strongly suggested authors should add the evaluations on leukaemia cell line(s) to compare the efficacies of these four metal complexes?
2. SRB method was mainly used for anti-proliferative evaluation of compounds while CCK8 method was an acknowledged method to evaluate the cytotoxicity. Therefore, I strongly suggested authors should add the CCK8 evaluation results.
3. I suggeted authors add the x-ray crystal structure of one of the metal complexes.
4. I suggeted authors showed the microscopical results of cellular uptake assay.
The Quality of English Language can be improved
Round 2
Reviewer 2 Report
1. I still persisted authors should put the x-ray crystal structure of target complexes, as long as even one complex. Because the structure and configuration is very important to explain the structure-activity relationship of these complexes.
2. Authors should present the microscopical results of cellular uptake assay in addition to the table.
Minor revision should be made to make the manuscript more readable
